# Almond By-Products Substrates as Sustainable Amendments for Green Bean Cultivation

**DOI:** 10.3390/plants13040540

**Published:** 2024-02-16

**Authors:** Vânia Silva, Ivo Oliveira, José Alberto Pereira, Berta Gonçalves

**Affiliations:** 1Centre for the Research and Technology of Agro-Environmental and Biological Sciences (CITAB), Institute for Innovation, Capacity Building and Sustainability of Agri-Food Production (Inov4Agro), University of Trás-os-Montes e Alto Douro (UTAD), Quinta de Prados, 5000-801 Vila Real, Portugal; ivo.vaz.oliveira@utad.pt (I.O.); bertag@utad.pt (B.G.); 2Centro de Investigação de Montanha (CIMO), Instituto Politécnico de Bragança, Campus de Santa Apolónia, 5300-253 Bragança, Portugal; jpereira@ipb.pt; 3Laboratório Associado para a Sustentabilidade e Tecnologia em Regiões de Montanha (SusTEC), Instituto Politécnico de Bragança, Campus de Santa Apolónia, 5300-253 Bragança, Portugal

**Keywords:** almond by-products, almond hulls, almond shells, antioxidant capacity, *Phaseolus vulgaris* L., quality, substrates, total phenolics

## Abstract

Almond processing generates a high quantity of by-products, presenting the untapped potential for alternative applications and improved sustainability in production. This study aimed to evaluate whether the incorporation of almond by-products (hulls/shells) can improve the biochemical characteristics of green bean pods when used as an alternative to traditional growing media in green bean plants. Four substrates were prepared: the Control substrate (C): 70% peat + 30% perlite; substrate (AS): 70% peat + 30% shells; substrate (AH): 70% peat + 30% perlite + 1 cm hulls as mulch; substrate (MIX): 70% peat + 15% shells + 15% hulls. Plants were grown in each of these substrates and subjected to two irrigation levels, 100% and 50% of their water-holding capacity. Biochemical parameters (photosynthetic pigments, total phenolics, flavonoids, *ortho*-diphenols, soluble proteins, antioxidant capacity) and color were evaluated in the harvested pods. Results showed that pods from plants growing in AH substrate presented statistically significant higher values in their total phenolic content, while AS and MIX substrates did not reveal significant benefits. Summarily, this study highlights the potential of almond hulls as a promising medium for green bean cultivation, particularly when employed as mulch. Further research is recommended to gain a more comprehensive understanding of the application of almond by-products as natural fertilizers/mulch.

## 1. Introduction

The global almond crop has shown steady growth, particularly in temperate regions, with a reported world almond production (in shell) estimated at 4 million tons in 2021 [1]. In the production and industrial processing of almonds, the greatest value is attributed to the edible portion, the kernel, while the remaining parts, namely the by-products, are typically discarded or undervalued [2]. These by-products, which include hulls, shells, and skins (Figure 1), constitute a significant proportion of the total production volume. In terms of percentage, the hulls account for 52% of the total fresh weight of the almond fruit, the shells represent 33%, the kernel with the skin makes up 15% [3], and the skin alone represents about 4% [4,5].

Traditionally, almond hulls, the grey-green outer shell that surrounds the inner brown hard shell, have conventionally served as livestock feed and fuel [3,5,6]. Almond shells, representing the intermediate layer between the hull and the kernel, exhibit structural rigidity owing to their fibrous composition, primarily cellulose, hemicelluloses, and lignin [7,8]. These shells have been employed mainly in the production of coal, fuel, furfural, xylose, and xylitol, but their versatility is further exemplified by their application as dietary antioxidants, as mulch in gardens and growing substrates, as an adsorbent for heavy metals or dyes, or in the preparation of activated carbons [5,9,10]. Almond tegument, also known as a pellicle, seed coat, or almond skin, is generated as a by-product during the industrial blanching and peeling process. These skins are a valuable agricultural by-product widely reported as a good source of phenolic compounds [5,11,12,13], and their high antioxidant power allows them a wide range of applicability [11,12,13].

Seen for a long time as waste, agri-food by-products present themselves as valuable resources with high potential. In recent years, several authors highlighted the physicochemical attributes of these by-products, such as nutritional, functional, and bioactive components, which have a huge potential to be used [14,15,16,17,18,19,20,21,22]. Therefore, it is imperative to explore alternative applications to minimize waste and advocate for more sustainable production practices. The involvement of the scientific and industrial community, as well as political decision-makers, is crucial in this endeavor.

Traditional substrates, which typically consist of materials like peat, sand, and perlite, are widely employed in vegetable crops as a substitute for conventional greenhouse agricultural soil [23,24]. Currently, peat is one of the organic materials most used as a substrate for the soilless cultivation of horticultural crops, being an important component of containerized mixtures for commercial plant production [25]. To mitigate expenses and negative environmental consequences associated with traditional substrates, including peat, various researchers have studied the potential utilization of organic residues and agricultural by-products as alternatives in soilless cultivation systems [26,27,28,29,30,31]. Some studies have explored the potential of incorporating almond by-products into substrates for potted ornamental and horticultural plants [32,33], as a natural soil fertilizer and as mulch in *Phaseolus vulgaris* plants [3,7,34,35]. Nevertheless, there remain significant uncharted possibilities in this field. Considering almond hulls and shells as perlite alternatives in greenhouse crop substrates holds promise as a sustainable option worth investigating.

In the marginal tropics, the production of *P. vulgaris* typically occurs in rainfed systems, which often yield low harvests, mainly due to the adverse effects of drought and limited availability of phosphorus (P) in the soil [36]. The majority of production losses arise from abiotic stresses that impact plants in the field, encompassing factors such as drought, soil salinity, high temperatures, cold, ultraviolet light, or flooding. Among these, drought is the most limiting environmental stress condition for agriculture. Given that 60% of this species production takes place on agricultural land susceptible to water deficit, it becomes important to investigate techniques that enhance the resilience of beans against drought [37,38,39].

Nowadays, the pressing issues of climate change and food shortages affect populations globally. In response to these challenges, there is a growing urgency to reduce waste and explore sustainable alternatives in food production. These alternatives should not only minimize waste but also address the escalating impact of severe drought periods.

Considering the significant expansion of global almond production and the substantial quantities of by-products generated during its production and industrial processing, a significant portion of which is currently discarded, it is imperative to urgently explore sustainable alternatives for their utilization.

Hence, the primary aim of this study was to assess the feasibility of utilizing almond by-products (specifically, almond hulls—AH and almond shells—AS) in growth chambers as potential substrates for plant cultivation. Thus, an experiment was conducted on green bean plants, encompassing a biochemical characterization along with the evaluation of the antioxidant capacity (AC) of the produced pods. Two irrigation levels were also tested: irrigation at 100% and at 50% of the water-holding capacity (WHC) of each substrate. The WHC is the amount of water that remains in a container after drainage stops due to saturation [40]. Therefore, the distinct substrates will be designated by their initials C, AH, AS, and MIX according to the by-product to which they refer, followed by the number 100 in the case of 100% and the number 50 in the case of 50% of WHC.

## 2. Results and Discussion

### 2.1. Substrate Characteristics—pH, Temperature, Relative Humidity, and Electrical Conductivity

The pH, temperature (T), relative humidity (RH), and electrical conductivity (EC) values of the distinct substrates are shown in Figure 2. The results suggest that the type of substrate, the irrigation level, as well as the interaction between both factors influenced both the pH and the RH obtained (*p*(S) < 0.001, *p*(R) < 0.001, *p*(S×R) < 0.001). It was found that the pH varied between 5.80 (MIX50) and 7.10 (AH50). The pH of the substrates C, AH, and AS remained neutral, with slight fluctuations, throughout the trial, while the MIX substrates showed a slight acidity but very close to neutrality as well. Thus, in terms of pH, these substrates proved to be suitable for plant growth. In relation to RH, substrates AH and AS presented lower values (on average 58%) compared to substrates C (on average 66%) and MIX100 and MIX50 (on average, 58% and 62%, respectively). The ideal soil moisture content for crops varies based on the particular plant species, with the generally recommended range falling between 20% and 60%, while vegetables should have soil moisture between 41% and 80% [41]. In high-humidity conditions, soil has a tendency to retain more moisture, fostering increased microbial activity and nutrient availability. However, excessive humidity can also result in waterlogging and reduced oxygen availability in the soil, which may have adverse effects on plant growth [41]. Thus, the RH values obtained were close to the ideal range of values for crops. Electrical conductivity (EC, mS/cm) measures the material’s ability to conduct an electrical current. Regarding the results of the EC, only the effect of the substrate was significant (*p*(S) < 0.001), and it was notable that MIX100 and MIX50 showed an EC much higher than the other substrates right at the beginning of the study varying between 1.47 mS/cm (MIX100) (corresponding to a combined total dissolved solids (TDS) in water of 1029 ppm), and 2.82 mS/cm (MIX50) (corresponding to a combined TDS in water of 1974 ppm). Since EC has an intimate relationship with the water and mineral content of soil, elements that determine the productivity of a crop, and that conductivity increases as salinity increases, it means that the MIX substrate offers greater availability of dissolved salts for the plant, proving to be, from the outset, a promising substrate for plant cultivation. Therefore, since the composition of the MIX substrate is a mixture of hulls and shells, this result reinforces previous theories that the use of almond shells increases the EC of substrates and nutrient solutions [7,42,43]. In relation to T, it was influenced by the type of substrate and differentiated irrigation (*p*(S) < 0.01, *p*(R) < 0.001). In legume cultivation, such as beans, the ideal temperature varies between 15 °C and 25 °C [44]. According to the results obtained, the substrates C100, AH100, and MIX100 presented an average T of 19 °C, the subtracts C50, AS100, and MIX50 an average T of 20 °C, and the substrates AS50 and MIX50 an average T of 21 °C. Therefore, all substrates presented optimal temperatures for the development of the culture under study [44].

### 2.2. Chromatic Parameters

The chromatic parameters of pods resulting from green beans grown in four distinct substrates are shown in Figure 3. The results showed that all the parameters evaluated revealed statistically significant differences among substrates. In terms of lightness (*L**) (Figure 3A), the highest value was recorded in pods from the C50 substrate (61.756 ± 5.247) accompanied, in the same order of magnitude, by C100, AH100, AH50, and AS50, and the lowest value in pods from MIX100 (41.305 ± 16.136). Regarding the chromatic coordinate a (*a**) (Figure 3B), substrates MIX100 (−6.435 ± 2.591) and MIX50 (−5.938 ± 3.603) stood out negatively, which presented lower values (fewer green pods) compared to the other substrates, including C100, which presented the highest value (−9.242 ± 0.819, greenest pods). The chromatic coordinate b (*b**) (Figure 3C) showed significant differences between the pods of the MIX50 (24.159 ± 9.132), the AH100 (30.087 ± 4.971), and the AS50 (29.173 ± 3.685) substrates. In terms of chroma (*C**) (Figure 3D), the lowest value was in pods from the MIX100 substrate (25.144 ± 9.063) compared to those from other substrates, but mainly with those from the AH100 substrate, which presented a higher value (31.238 ± 4.581). Regarding the hue (Figure 3E), all pods from all substrates displayed a green color (same range of values, around 178°). The effect of the substrate was noted on *L** and *a** (*p*(S) < 0.001) and also on *C** and *h*° (*p*(S) < 0.05). The effect of the irrigation was observed in *b** and *C** (*p*(R) < 0.05). On the other hand, the differences observed in *a**, *b**, and *h*° were also due to the effect of the interaction between the substrate and irrigation (*p*(S×R) < 0.05 for *a**, *b**, and *p*(S×R) < 0.001 for *h*°).

In comparison to Proulx et al. (2010) [45], who reported a lightness range of values between 47 and 56 and a chroma of 19 to 28 for two varieties of snap beans—approximately aligning with our observed range (24–30 for *C** and 40–60 for *L**). However, our hue results surpassed theirs, registering 178 compared to their reported 121–123. In another study, Kasim and Kasim (2015) [46] obtained lightness values of 65.93, a chromatic coordinate *a* of around −14, and a hue of around 115.5 for samples of fresh-cut green beans. Lightness has been used by several researchers as an indicator of vegetable spoilage [47]. In our study, a decrease in *L** values was noted in the MIX substrates and also in AS100, which might suggest that the pods from these substrates exhibited signs of deterioration.

### 2.3. Quantifications in Green Bean Pods

#### 2.3.1. Photosynthetic Pigments

The photosynthetic pigment pods resulting from green beans grown in four distinct substrates are shown in Figure 4. The results pointed to significant differences only in the value of the chlorophyll/carotenoids ratio (Total (Chl/Carot.)), highlighting the highest value in the C substrates (4.149 ± 0.020 mg g^−1^ FW and 4.174 ± 0.017 mg g^−1^ FW for C100 and C50, respectively) (Figure 4F). These differences were attributed to the substrate effect (*p*(S) < 0.001). This is contrary to the results obtained by Oliveira et al. (2019) [35], who obtained significant differences in the values of photosynthetic pigments presented by green bean pods from distinct substrates. However, the results obtained in our study were notably higher than those obtained by these authors for all types of pigments evaluated. Oliveira et al. (2019) [35] highlighted that the mixture of peat and almond shells (80:20) increased the carotenoid content in the harvested pods, similar to what happened with Oruña-Concha et al. (1997) [48]. Our values are considerably lower than those reported by El-Nafad et al. (2022) [49], who obtained a range of values between 3.88–4.11 mg g^−1^ DW to Chl *a* content, 2.56–2.75 mg g^−1^ DW to Chl *b*, and 1.13–1.60 mg g^−1^ DW to total carotenoids, in conventional and organic fresh green beans, respectively. Contradictory to what would be expected, the values of Chl *b* were higher than those of Chl *a*, which may indicate that the plants were under some type of stress, such as being shaded (with insufficient light). Some studies associate an increase in chlorophyll *b* with shaded plants [50,51,52,53]. Muhidin et al. (2018) [54], for example, found that chlorophyll *b* increased in several varieties of shaded plants compared to plants more exposed to light. In shaded plants, it increases the wavelengths that can be captured by the plant. Thus, the increased proportion of chlorophyll *b* in shade plants is due to its absorption properties [55]. Green beans are considered a crop with high thermal and light requirements [56,57]. The effect of reduced irradiance (shading) on common bean is characterized by several adverse effects, such as a decrease in the yield and plant biomass proportional to the reduction of solar light and a decrease in the mean grain and shoot weight [58,59], lower chlorophyll content per unit area, a reduced Chl (*a*/*b*) ratio, low stomatal density, and a reduced net assimilation rate [60]. This fact may indicate that the radiation of 300 μmol m^−2^ s^−1^ was insufficient for this crop. Of course, biological systems do not always respond to conditions in a similar fashion, depending on the type of plant life stage and nutritional conditions.

#### 2.3.2. Total Phenolics, Flavonoids, *Ortho*-Diphenols, and Soluble Proteins

The biochemical parameters of pods resulting from green beans grown in four distinct substrates are shown in Figure 5. According to the results of the pods’ biochemistry, it was possible to verify that all the parameters evaluated revealed statistically significant differences among substrates. In Figure 5A, it was found that the differences observed in the phenolic content were influenced either by the isolated effect of the substrate (*p*(S) < 0.001), or by irrigation (*p*(R) < 0.001), or by the interaction between both (*p*(S×R) < 0.001). It was also observed that the pods from the AH substrate presented a higher total phenolic content (2.160 ± 0.056 mg GAE g^−1^ FW), with the results being statistically more significant in the pods from plants irrigated at 100% WHC, compared to the Control and with the other substrates. The pods from plants in the AS substrate irrigated at 50% WHC were those with the lowest total phenolic content (0.357 ± 0.010 mg GAE g^−1^ FW). It should be noted that substrates AH50 and AS100 presented values in the same order of magnitude as substrates C100 and C50, which demonstrated that the total phenolic content of the pods resulting from plants that grew in these substrates was not affected. Overall, our values fit the range observed by Carbas et al. (2020) [61] (0.11–4.59 mg g^−1^ DW) and by Mastura et al. (2017) [62] (1.36–4.54 mg g^−1^ DW), even though lower (0.84–1.00 mg g^−1^ FW) [35], and higher (3.55 mg g^−1^ DW) [63], (6–8 mg g^−1^ DW) [64], (2.34–2.80 mg g^−1^ DW) [49] values have been recorded in other works. It should be noted that the differences in the results of the different authors may have to do with the different extraction and quantification methods that may interfere with the results obtained.

Regarding flavonoids (Figure 5B), there were no significant differences compared to the Control, with the exception of the pods of MIX100 (1.968 ± 0.084 mg CATE g^−1^ FW), which presented a lower concentration of flavonoids in comparison to the C100 pods (2.417 ± 0.254 mg CATE g^−1^ FW). The differences observed were due to the effect of the substrate (*p*(S) < 0.01). These results are in line with the range of values reported by Carbas et al. (2020) [61] (0.80–4.33 mg g^−1^ DW) and higher than the obtained by El-Nafad 2022 [49] (0.33–0.37 mg g^−1^ DW). In relation to *ortho*-diphenols (Figure 5C), the pods from plants in the AS50 and MIX100 substrates stood out negatively, presenting the lowest values (0.578 ± 0.039 mg GAE g^−1^ FW and 0.677 ± 0.222 mg GAE g^−1^ FW, respectively), with differences resulting of the use of distinct substrates (*p*(S) < 0.001). These results fit the lower range of values reported by Carbas et al. (2020) [61], who reported a content of *ortho*-diphenols that varied between 0.89–6.69 mg g^−1^ DW. Looking at Figure 5D, it can be seen that the lowest protein content was obtained in pods from the MIX100 substrate (3.439 ± 0.001 mg BSA g^−1^ FW), but the substrates AH100 (5.609 ± 0.977 mg BSA g^−1^ FW), AH50 (5.846 ± 0.360 mg BSA g^−1^ FW), and MIX50 (4.994 ± 0.621 mg BSA g^−1^ FW) presented values comparable to the C100 (5.926 ± 0.152 mg BSA g^−1^ FW) and to C50 (5.747 ± 0.654 mg BSA g^−1^ FW). All the values align with those reported by Oliveira et al. (2019) [35]. Nevertheless, unlike this previous study, which did not identify significant differences in the protein content of green bean pods across various substrates, the current research suggests a certain substrate influence (*p*(S) < 0.01). On the other hand, the values we obtained are lower than those found by Sánchez-Mata et al. (2003) [65], which were 16.4 mg g^−1^ DW.

### 2.4. Antioxidant Capacity (AC)

The antioxidant capacity (AC by ABTS and DPPH methods) of pods resulting from green beans grown in four distinct substrates is shown in Figure 6. The results pointed to a significant influence of the substrate (*p*(S)_ABTS, DPPH_ < 0.001) on the AC of the pods for both methods (ABTS and DPPH). Using the ABTS method, there was also a significant influence of irrigation (*p*(R)_ABTS_ < 0.001) and the interaction between the substrate and irrigation (*p*(S×R)_ABTS_ < 0.001) for both methods; higher AC was recorded in the C100 substrates, while for the ABTS method, lowest values were found in pods from the AH100 and using the DPPH method, lower values were recorded in pods from the MIX50 substrate. Oliveira et al. (2019) [35] did not obtain significant differences between substrates, with regard to the AC of the pods, by DPPH and ABTS methods, which varied between 4.21–4.46 µmol TE g^−1^ FW, in the pods from C substrate (80% peat:20% vermiculite), and in the pods from AM substrate (control + shell mulch) for AC-DPPH, respectively; and between 14.62–15.71 µmol TE g^−1^ FW, in the pods from C substrate, and in the pods from AS substrate (80% peat:20% shells) for AC-ABTS, respectively. Comparing our results with those obtained by Oliveira et al. (2019) [35], our results were superior compared to AC-DPPH and lower compared to AC-ABTS. Mastura et al. (2017) [62] reported that the AC of raw green beans ranged from 5.94–6.45 µmol TE g^−1^ DW by the DPPH method and from 17.12–22.29 µmol TE g^−1^ DW by the ABTS method, for organic and inorganic green beans, respectively.

To understand which compound(s) (phenolics, flavonoids, or *ortho*-diphenols) contributed most to the AC presented by the pods, we investigated how it varies depending on the concentration of each of these compounds and; for this purpose, linear regressions (Figure 7) were carried out. It was found that the linear relations are very low. A medium correlation between AC and flavonoid content, by the DPPH method, was highlighted (R^2^ = 0.536, *p* < 0.001).

Therefore, a Pearson correlation was also performed with the aim of verifying the associations/correlations between two variables involving all parameters under study.

### 2.5. Pearson Correlation for All Evaluated Parameters

The Pearson correlation matrix between all parameters under study is shown in Appendix A.

By observing the results of the correlation matrix (Appendix A), the *ortho*-diphenols are moderately related to Chl *b* (0.425*) and to Total Carot. (0.411*), and to phenolic compounds (0.577**). The flavonoids are moderately related to Chl *b* (0.425*) and to Total Carot. (0.420*), and to phenolics (0.427*), and strongly related to *ortho*-diphenols (0.714**). With regard to AC-ABTS, there was a strong and positive correlation to Total Carot. (0.805**), and also a strong but negative correlation to Total (Chl/Carot.) ratio (−0.895**), probably associated with the presence of β-carotene, which is considered a strong antioxidant. Given that carotenoids are pigments that play an important role in protecting plants against photooxidative processes [66], their presence may suggest that plants are undergoing photo-oxidation. Still, in relation to AC, there was also a moderate correlation with phenolics (0.426*) and *ortho*-diphenols (0.582**) for the DPPH method and also a strong correlation with flavonoids (0.732**). For proteins, it established a moderate correlation with Chl *b* (0.532**), phenolics (0.549**), flavonoids (0.566**), and AC-DPPH (0.582**), and a strong correlation to *ortho*-diphenols (0.835**). This strong correlation has to do with the fact that plant phenolics have the capability to bind either covalently or non-covalently to proteins, with the nature of these interactions depending on the mole ratio of phenolics to proteins [67,68].

### 2.6. Principal Component Analysis (PCA)

Principal Component Analysis (PCA) uses a correlation matrix to standardize the data (Corr-PCA).

In order to enhance the comprehension of the interrelationships among the 12 assessed parameters of pods resulting from green beans grown in four distinct substrates, irrigated at 100% and 50% of WFC, a comprehensive chemometric analysis was conducted, integrating all the data. The analysis revealed that the biochemical composition of these pods could be explained by three principal components (Figure 8). To assess the reliability of the PCA, the Kaiser–Meyer–Olkin (KMO) and Bartlett tests were performed, as well as a Varimax rotation. The eigenvalues of the three principal components surpassed one and cumulatively explained 81.81% of the total variance in the dataset, with factor 1 bearing the greater weight at 34.71% (Table 1). Specifically, the first principal component (PC1) explained 34.71% of the total variance, indicating high loadings for phenolics, *ortho*-diphenols, flavonoids, antioxidant capacity by the DPPH method, and proteins; the second principal component (PC2) which accounted for 28.38% indicated high values for chlorophyll *a*, chlorophyll *b*, total chlorophyll, the ratio of total chlorophyll/carotenoids, and the ratio of chlorophyll (*a*/*b*); in turn, the third principal component (PC3) explained 18.73% indicated high values for total carotenoids as well as antioxidant capacity by the ABTS method.

## 3. Materials and Methods

### 3.1. Plant Material

#### 3.1.1. Green Bean Seeds

For this trial, green bean seeds of the ‘Bencanta’ climbing bean variety were used. ‘Bencanta’ is a traditional variety from the Beiras region, registered in the national catalog of Portuguese varieties with the ‘Associação Nacional dos Produtores e Comerciantes de Sementes—ANSEME’ as its proponent or maintainer [69]. ‘Bencanta’ is also registered on the EU database of registered plant varieties as PT b 9 and as belonging to the species *Phaseolus vulgaris* L.—27.2. Climbing French Bean Group [70].

Its production is in large bunches, very tender and with an excellent flavor, and its red grain is streaked with white early.

#### 3.1.2. Almond Shell and Almond Hulls

The almond by-products (shells and hulls) used in this study came from an almond orchard around 15 years old installed in Lamas de Orelhão, Mirandela (Trás-os-Montes region, North of Portugal), and relate to varieties ‘Ferraduel’, ‘Ferragnès’, ‘Marinada’, and ‘Lauranne’ from the 2021 harvest. After harvesting, the almond shells were manually separated and dried in an oven at 60 °C, and the shelled almonds were air-dried at room temperature (25–30 °C). Subsequently, the shells were separated from the almond kernels using a manual nutcracker. Next, the cracked almond shells and hulls were sieved (0.5–1 cm) to ensure that the particles were homogeneous and did not exceed approximately 1 cm.

### 3.2. Almond Shell and Almond Hull Substrates

In order to evaluate the effect of almond hulls and shells on green bean cultivation, four distinct substrates were prepared, composed of different volumetric proportions of these two by-products, and without further processing. The control substrate (C) was a mixture of 70% peat and 30% perlite. The almond shell substrate (AS) was made of 70% peat and 30% shells (in order to maintain the same control concentrations but replace the perlite with shells). The almond hulls substrate (AH) was a mixture of 70% peat and 30% perlite covered with 1 cm of almond hulls as mulch, and a mixed substrate (MIX) composed of a mixture of 70% peat, 15% shells, and 15% hulls (in order to maintain the same control concentrations but replacing the perlite with a mixture of shells and hulls).

### 3.3. Growth Conditions and Experimental Design

The assay was performed as a completely randomized design with three replicates for treatment encompassing 4 types of substrates (C, AH, AS, and MIX) with 2 irrigation levels each, 100% and 50% of water-holding capacity (WHC). The treatments were designated as C100, C50, AH100, AH50, AS100, AH50, and MIX100, MIX50. We used 3 pots per treatment and 3 plants per pot, totaling 72 plants under study (24 pots) (Figure 9).

Previously, green bean seeds were placed to germinate in an alveolar tray and then, about 2 weeks later, transplanted into pots with a capacity of 3 L, with the respective substrates under study. All transplanted plants were homogeneous and had at least 10 cm, 2 primary leaves and 3 tiny definitive leaves.

The pots were randomly placed in the growth chamber (FitoClima 10,000 EHHF, Aralab, Rio de Mouro, Portugal) under controlled climate conditions: a 16 h light period at 23 °C and an 8 h dark period at 18 °C, with a photosynthetic photon flux density of 300 μmol m^−2^ s^−1^. Relative humidity was maintained at 75% and 80% during the light and night periods, respectively.

Before starting the differentiated irrigation, plants were uniformly irrigated, with each pot receiving 500 mL of tap water once a week.

The differentiated irrigation began around 2 weeks after transplanting into a pot when it seemed that the plants were already well adapted to the substrate. Thus, plants were irrigated once a week at two different irrigation conditions, 50% and 100% of WHC. Four rounds of irrigation were also performed with Hoagland’s nutrient solution (100 mL/pot) [71]. The initial irrigation took place one week after transplanting into a pot, followed by subsequent irrigations every two weeks.

After 60 days of growth in a pot, the assay was finished, and the pods visually considered suitable for consumption were harvested, and the others were considered waste.

The harvested pods were properly identified, the color was measured in each one and then deep-frozen in liquid nitrogen and stored at −80 °C for later analysis of photosynthetic pigments (chlorophylls and carotenoids), soluble proteins, total phenolics, flavonoids, and *ortho*-diphenols, and also the antioxidant capacity. When used for the different quantifications, the previously deep-frozen pods (including seeds) were macerated in liquid nitrogen, and the respective mass required for each extraction/quantification was weighed.

### 3.4. Substrate Characteristics—pH, Temperature, Relative Humidity, and Electrical Conductivity

The evolution of the pH, temperature (T), relative humidity (RH), and electrical conductivity (EC) values of the distinct substrates was determined and registered weekly before irrigating the plants using a portable digital meter. For EC soil determination was used a digital EC/Temperature Soil Tester 0.00~10.00 mS/cm, EC-8801, Nenninger GmbH, Berlin, Germany; and for the pH, T, and RH determinations was used a multifunctional soil detector, 5 in 1 potted pot soil tester, Soonda, Beijing, China.

### 3.5. Chromatic Parameters

The color of green bean pods was measured on both sides, according to the CIELAB color space system of 1976 (CIE, Commission International de l’Eclairage), using a colorimeter (CR-300, Minolta, Osaka, Japan). The CIELAB color space covers the range of human color perception and precisely distinguishes color differences using three color values as a measurement, which are represented by the letters *L** (lightness), *a** (chromatic coordinates *a**) and *b** (chromatic coordinates *b**). *L** represents lightness from black (or opaque) to white (or transparent) and can be read on a scale of 0 to 100. Chromatic coordinates *a** and *b** represent chromaticity (saturation) with no specific numeric limits, while coordinate *a** can be read from red to green (+*a** corresponds with red and −*a** corresponds with green), coordinate *b** can be read from yellow to blue (+*b** corresponds with yellow and −*b** corresponds with blue). The color can also be described using the Munsell color system, which is a color space that specifies colors based on three color proprieties represented by three cylindrical coordinates: value or lightness (*L**), color intensity or chroma (*C**), and basic color or hue (*h*°) [72,73]. The color nuance can vary between 0° to 90° (red to yellow) and 180° to 270° (green to blue-green) and is obtained according to the following formula: hue = arctg (*b**/*a**) [74,75]. The color intensity (*C**) of green bean pods was accessed according to the formula: *C** = (*a**^2^ + *b**^2^)^1/2^ [76,77,78]. Results were presented as the average of the total harvested pods for each substrate with the indication of standard deviation (SD).

### 3.6. Quantifications in Green Bean Pods

#### 3.6.1. Photosynthetic Pigments

The quantification of photosynthetic pigments (chlorophyll *a* (Chl *a*), chlorophyll *b* (Chl *b*), and carotenoids) followed a procedure adapted from Arnon (1949) [79] and Lichtenthaler (1987) [80].

Extractions were performed on 25 mg of sample (FW), which had been previously macerated in liquid nitrogen, in a total volume of 4 mL of acetone/distilled water, 80:20 (*v*/*v*). The samples were vortexed and centrifuged at 4000 rpm for 10 min at 4 °C, and the supernatant was collected. For each sample, 200 μL of extract was pipetted into a 96-well microplate, and the absorbances were read in a microplate reader at wavelengths (λ) 663, 645, and 470 nm.

The procedure was always carried out, protected from light, and with the samples kept in the cold.

The calculation of photosynthetic pigments and carotenoids (mg mL^−1^) was performed using the following formulas:Chlorophyll *a*: Chl *a* = ((12.7 × Abs663) − (2.69 × Abs645))/1000;
Chlorophyll *b*: Chl *b* = ((22.9 × Abs645) − (4.68 × Abs663))/1000;
Total Chlorophyll: Total Chl = ((20.2 × Abs645) + (8.02 × Abs663))/1000;
 Total Carotenoids: Total Carot. = (((1000 × Abs470) − (1.82 × Chl *a*) − (85.02 × Chl *b*))/198)/1000

#### 3.6.2. Total Phenolics, Flavonoids, and *Ortho*-Diphenols

Total phenolic compounds, flavonoids, and *ortho*-diphenols were quantified in previously prepared methanolic extracts, each using different methods.

The sample extraction procedure involved adding 1.5 mL of methanol/distilled water (70:30, *v*/*v*) to 40 mg of fresh green bean pod material, thoroughly mixing in a vortex. Subsequently, the mixture was agitated for 30 min in the dark, followed by centrifugation at 10,000 rpm for 15 min at 4 °C. This extraction was repeated three times, and supernatants from successive extractions of each sample were combined, resulting in a final volume of 5 mL filled with the aforementioned solvent. The resulting extracts were stored at −20 °C until subsequent analysis.

Total phenolics

To quantify total phenolics, the Folin–Ciocalteu colorimetric method was used, a procedure adapted by Singleton and Rossi (1965) [81] and Dewanto et al. (2002) [82], with some modifications. Thus, 20 μL of extract, 100 μL of Folin–Ciocalteu reagent (1:10), and 80 μL of sodium carbonate (Na_2_CO_3_, 7.5%) were added to a 96-well microplate and the absorbances were read (λ = 765 nm). Gallic acid (100 mM) was used as a standard to determine the concentration of phenolic compounds. Results were presented as milligrams of gallic acid equivalents (GAE) per gram of fresh weight (mg GAE g^−1^ FW);

2.Flavonoids

The flavonoid’s quantification was performed according to Dewanto et al. (2002) [82] by adding, to each well of the 96-well microplate, 25 µL of extract + 100 µL of bi-distilled water + 10 µL of NaNO_2_ (5%), after 5 min at room temperature, another 15 µL of AlCl_3_ (10%) was added and, after another 6 min, more than 100 µL of NaOH (1 M) + 50 µL of bi-distilled water. The absorbances were read in a microplate reader (λ = 510 nm). Catechin (Sigma-Aldrich Chemie GmbH, Taufkirchen, Germany) (1 mg mL^−1^) was used as a standard to determine the concentration of flavonoids. Results were presented as milligrams of catechin equivalents (CATE) per gram of fresh weight (mg CATE g^−1^ FW);

3.*Ortho*-diphenols

To determine the *ortho*-diphenols content, 160 µL of each extract and 40 µL of sodium molybdate (Na_2_MoO_4_2H_2_O, 5%) were added to each well of the 96-well microplate, and after incubation at room temperature, protected from light, for 15 min, absorbances were read (λ = 375 nm). Gallic acid (250 mg L^−1^) was used as a standard to determine the concentration of *ortho*-diphenols. Results were presented as milligrams of gallic acid equivalents (GAE) per gram of fresh weight (mg GAE g^−1^ FW). The procedure was adapted by Gouvinhas et al. (2018) [83] and Machado et al. (2017) [84], with minor adjustments.

#### 3.6.3. Soluble Proteins

Soluble proteins were quantified following the procedure adapted from Bradford (1976) [85]. The procedure consisted of the homogenization of 25 mg of sample with 1400 µL of freshly extraction medium containing 40 mL of phosphate buffer (50 mM, pH 7.5) with ethylenediaminetetraacetic acid—EDTA (0.1 mM) + 40 µL of phenylmethylsulfonyl fluoride—PMSF (100 mM) + 0.8 g of polyvinylpyrrolidone—PVP (2%, *w*/*v*), and posterior centrifugation at 12,000 rpm, 30 min, 4 °C. For quantified, 80 µL of extract and 200 µL of commercial Bradford reagent (Sigma-Aldrich Chemie GmbH, Taufkirchen, Germany) (200 µg mL^−1^) were added to each well of the 96-well microplate. The absorbances were read (λ = 595 nm) after 15 min in the dark, at room temperature. Bovine Serum Albumin (BSA, Sigma-Aldrich Chemie GmbH, Taufkirchen, Germany) (200 µg mL^−1^) was used as standard. The results were expressed as mg BSA per gram of fresh weight (mg BSA g^−1^ fresh weight (FW)).

### 3.7. Antioxidant Capacity

The antioxidant capacity was evaluated in the same methanolic extracts obtained for quantification of total phenolics, flavonoids, and *ortho*-diphenols. Two methodologies were used, namely the DPPH (2,2-diphenyl-1-picrylhydrazyl) radical scavenging activity and the scavenging effect of 2,2-azinobis-(3-ethylbenzothiazoline-6-sulfonic acid (ABTS^•+^)) radical.

1.DPPH method

The DPPH radical scavenging activity method is based on the reduction of DPPH by reaction with an antioxidant, resulting in a change in the initial color (purple) that gradually disappears, depending on the concentration of antioxidants present in the extract. This color change is accompanied by a decrease in absorbance. The procedure used was adapted from Siddhuraju and Becker (2003) [86], Sánchez-Moreno et al. (1998) [87], and Brand-Williams et al. (1995) [88] and consisted of mixing 15 µL of extract with 285 µL of DPPH (10^−5^ mol L^−1^) freshly working solution. This mixture was left in the dark, at room temperature for 30 min, and then the absorbance was read at a wavelength of 517 nm;

2.ABTS method

The ABTS^+^ method is based on the evidence of the antioxidant reaction with the ABTS^•+^ radical. In this reaction, there is a reduction of ABTS^+^ to ABTS, which is accompanied by a reduction in the initial color of ABTS^+^ that gradually disappears according to the concentration of antioxidants present in the sample. The procedure used was adapted from Stratil et al. (2006) [89] and consisted of mixing 15 µL of sample with 285 µL of ABTS working solution (previously prepared with 88 µL of K_2_S_2_O_8_ (140 mM) and 5 mL of ABTS (7 mM)). This mixture was left in the dark, at room temperature for 10 min, and then the absorbance was read at a wavelength of 734 nm.

For all methods, Trolox (Sigma-Aldrich Chemie GmbH, Taufkirchen, Germany) (3 mM) was used as a standard to determine the concentration of antioxidants present in the extracts.

All results were expressed as μmol of Trolox equivalent (TE) per gram of fresh weight (μmol TE g^−1^ FW).

### 3.8. Statistical Analysis

Results are presented as mean ± standard deviation (SD) of three replicates, presented by fresh weight (FW) of green bean pods. For color, results are presented as mean ± standard deviation (SD) of the total harvested pods (FW) for each treatment. For statistical analysis, the IBM SPSS Statistics 25.0 software (Statistical Package for Social Sciences, SPSS-IBM Corporation, Armonk, NY, USA) was employed. One-way analysis of variance (ANOVA) was used to detect differences among means. Comparison of means was performed using Tukey’s post-hoc multiple range test, with a significance level of 5% (*p* < 0.05).

Furthermore, Pearson’s rank correlation was employed to assess the correlation between antioxidant capacity and phenolic composition. A Principal Component Analysis (PCA) was also performed for exploratory data analysis and development of predictive models. The process involved the normalization of the data matrix for each attribute and the subsequent decomposition of the data correlation matrix (Corr-PCA) into eigenvalues.

## 4. Conclusions

Given the high production and resulting by-products of almonds, their use as alternative substrates should be easily available at reasonable prices and in sufficient quantities. Furthermore, these materials appear to be environmentally friendly, and their use would reduce waste resulting from the almond industry while also reducing the costs inherent to the use of perlite in substrates.

Since the physicochemical properties of substrates can potentially lead to nutritional deficiencies in crops, it is crucial to thoroughly test any new materials intended for use as growing media.

This study allowed us to assess the effect of distinct substrates, subject to two different levels of irrigation, on green bean pods. Summarily, and in terms of soil moisture, the results obtained point to the isolated effect of the substrate and the level of irrigation on the temperature, relative humidity, and pH and to the combined effect of the substrate and irrigation on pH and RH values. In general, the values obtained fall within the ideal range of values for vegetable cultivation, which suggests that our substrates are favorable to these crops. Regarding EC, there was only a significant effect of the substrate, with greater conductivity and, therefore, greater availability of dissolved salts for the plant, particularly notable in MIX substrates (based on almond hulls and shells), proving to be a promising substrate for plant cultivation. Thus, this result suggests that the combined use of almond hulls/shells increases the EC and the nutrient solutions of the substrates. Furthermore, the substrates based on almond by-products in the portions tested exhibited a favorable water retention capacity. This property leaves open the potential for its use as a substrate in plants susceptible to drought, as it would allow the plant to have water available for longer.

In terms of chromatic parameters and photosynthetic pigments, the use of substrates based on almond by-products does not appear to increase the color characteristics of the pods compared to traditional substrates (Control substrates).

Concerning the biochemistry of the pods, there were statistically significant differences between substrates in all parameters evaluated. The variations observed in phenolic content were influenced by either the isolated effect of the substrate or irrigation or by the interaction between the two. It should be noted that AH100 substrate presented a higher total phenolic content compared to the Control and the other substrates, and AH50 and AS100 substrates presented values in the same order of magnitude as the control substrates, which indicates that these substrates are able to maintain the availability of phenolics in relation to traditional substrates (Control substrates). The same happened with the content of proteins for substrates AH and MIX50. The effect of the substrate based on almond by-products did not prove to be promising with regard to the content of flavonoids (particularly in the pods from MIX100 compared to the pods from C100) nor in the content of *ortho*-diphenols (particularly in the pods from the substrates AS50 and MIX100).

Therefore, this study highlights the potential of almond hulls, without further processing, as a promising medium for green bean cultivation, particularly when employed as mulch, since its use not only did not negatively affect the characteristics of the pods but also increased their phenolic content. However, further research incorporating other substrate proportions, varied irrigation levels, or different crop species is recommended to gain a more comprehensive understanding of the application of almond by-products as natural fertilizers or mulches. Since peat is one of the most used substrates in nurseries, resulting from the exploitation of non-renewable resources that lead to the degradation of peatlands, causing environmental constraints, its availability may be in question in the near future. Thus, it will be relevant to study alternative substrates for their total or partial use. Therefore, it would be interesting to explore the possibility of using almond by-products in composting and/or vermicomposting for later use as an alternative substrate to peat in plant growth. Almond hulls and shells can be valuable in releasing their nutrients by the soil microbiome, as well as the many co-benefits for soil health, thus supporting the long-term sustainability of almond production systems.

Of course, to ensure the best growing conditions for seedlings, a thorough analysis of the specific substrates used should be conducted. And to make good use of the results obtained, they must always be replicated in agricultural practice through several stages in cooperation with horticultural companies.

## Figures and Tables

**Figure 1 plants-13-00540-f001:**
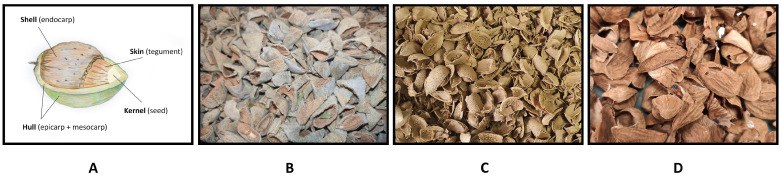
Parts of almond fruit (**A**) and almond by-products: hulls (**B**), shells (**C**), and skins (**D**).

**Figure 2 plants-13-00540-f002:**
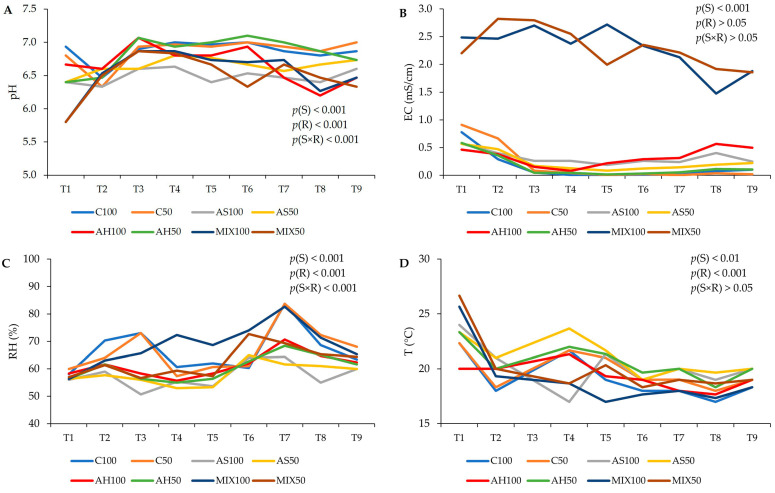
Evolution of the characteristics of distinct substrates, irrigated at 100% and 50% WHC, over the time the assay was carried out. (**A**) pH; (**B**) electrical conductivity (EC, mS/cm); (**C**) relative humidity (RH, %); and (**D**) temperature (T, °C). Abbreviations: T1—1st week in pot, T2—2nd week in pot, T3—3rd week in pot, T4—4th week in pot, T5—5th week in pot, T6—6th week in pot, T7—7th week in pot, T8—8th week in pot, T9—9th week in pot.

**Figure 3 plants-13-00540-f003:**
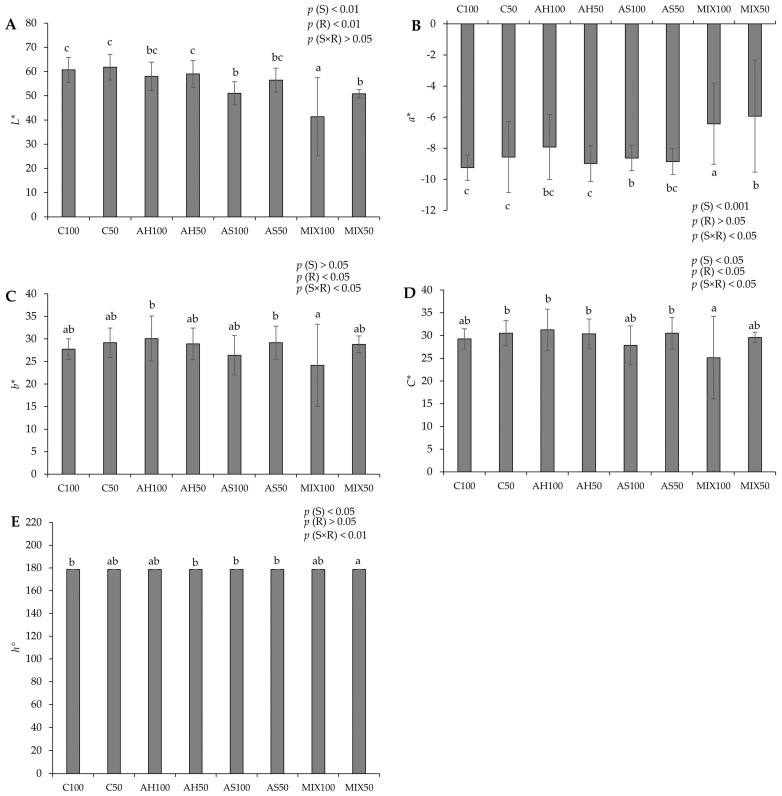
Chromatic parameters of pods resulting from green beans grown in four distinct substrates irrigated at 100% and 50% of WHC. (**A**) lightness (*L**); (**B**) chromatic coordinate *a* (*a**); (**C**) chromatic coordinate *b* (*b**); (**D**) chroma (*C**); (**E**) hue angle (*h*°). The values presented result from the means ± SD of the total number of pods harvested for each treatment. Statistically significant differences (*p* < 0.05) among substrates are marked with different letters, as determined by analysis of variance (ANOVA) and multiple comparisons test (Tukey test). Abbreviations: FW—fresh weight; (S) = substrate effect; (R) = irrigation effect; (S×R)= effect of the interaction between substrate and irrigation.

**Figure 4 plants-13-00540-f004:**
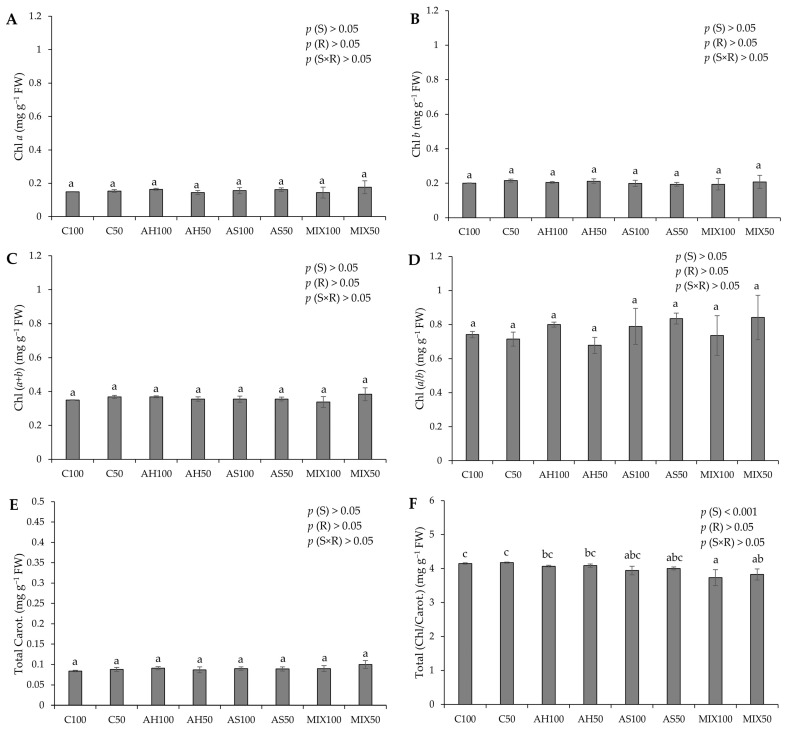
Photosynthetic pigments of pods resulting from green beans grown in four distinct substrates irrigated at 100% and 50% of WHC. (**A**) chlorophyll *a* (Chl *a*); (**B**) chlorophyll *b* (Chl *b*); (**C**) total chlorophyll content (Chl (*a* + *b*)); (**D**) Chl (*a*/*b)* ratio; (**E**) total carotenoids (Total Carot.); (**F**) total Chl/Carot. ratio. The values presented results from the means ± SD (n = 3). Statistically significant differences (*p* < 0.05) among substrates are marked with different letters, as determined by analysis of variance (ANOVA) and multiple comparisons test (Tukey test). Abbreviations: FW—fresh weight; (S) = substrate effect; (R) = irrigation effect; (S×R)= effect of the interaction between substrate and irrigation.

**Figure 5 plants-13-00540-f005:**
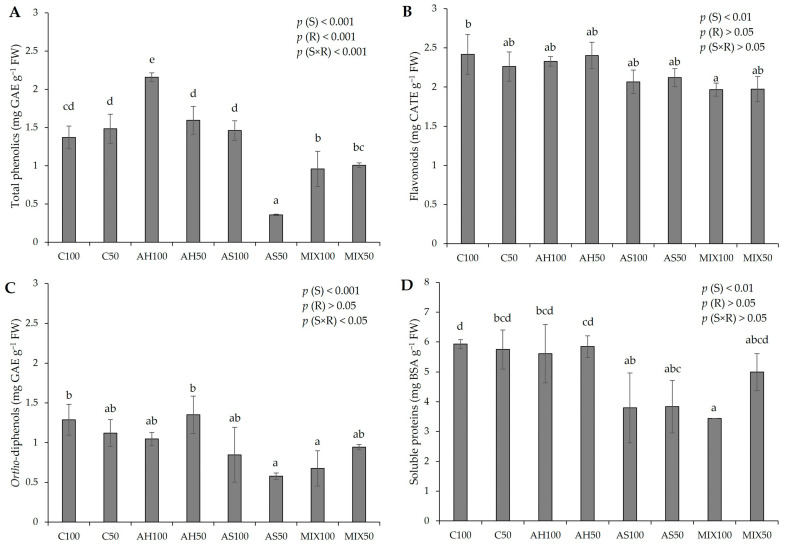
Biochemical parameters of pods resulting from green beans grown in four distinct substrates irrigated at 100% and 50% WHC. (**A**) total phenolics; (**B**) flavonoids; (**C**) *ortho*-diphenols; and (**D**) soluble proteins. The values presented are the result of the means ± SD (n = 3). Statistically significant differences (*p* < 0.05) among substrates are marked with different letters, as determined by analysis of variance (ANOVA) and multiple comparisons test (Tukey test). Abbreviations: GAE—gallic acid equivalents; CATE—catechin equivalents; BSA—bovine serum albumin; FW—fresh weight; (S) = substrate effect; (R) = irrigation effect; (S×R) = effect of the interaction between substrate and irrigation.

**Figure 6 plants-13-00540-f006:**
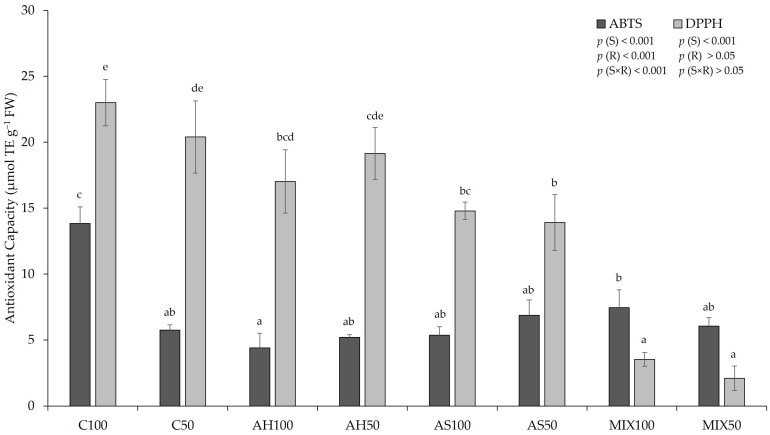
Antioxidant capacity (by the ABTS and DPPH methods) of pods resulting from green beans grown in four distinct substrates, irrigated at 100% and 50% of WHC. The values presented results from the means ± SD (n = 3). Statistically significant differences (*p* < 0.05) among substrates are marked with different letters, as determined by analysis of variance (ANOVA) and multiple comparisons test (Tukey test). Abbreviations: TE—trolox equivalents; FW—fresh weight; (S) = substrate effect; (R) = irrigation effect; (S×R) = effect of the interaction between substrate and irrigation.

**Figure 7 plants-13-00540-f007:**
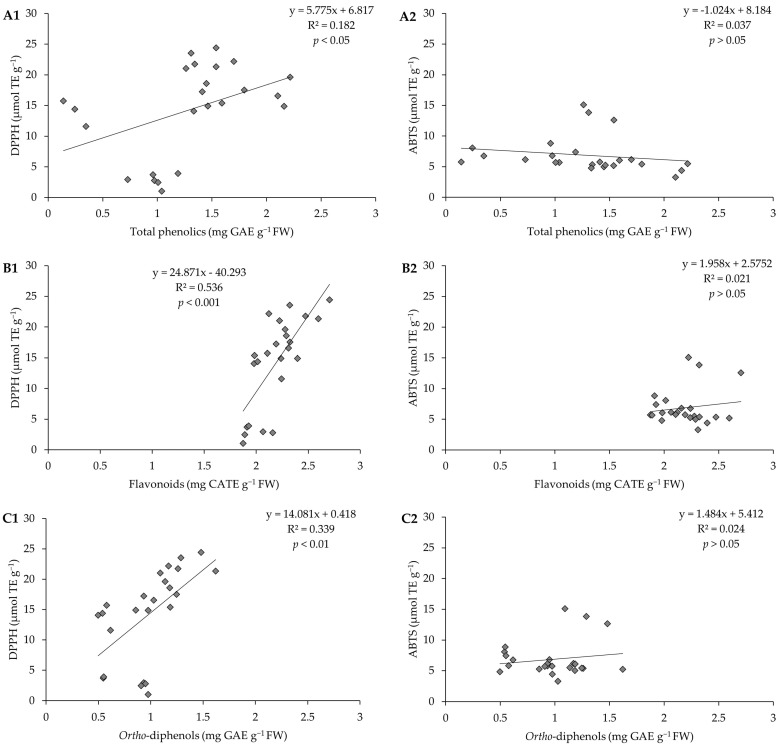
Correlation between (**A1**) antioxidant capacity by the DPPH method and total phenolics, (**A2**) antioxidant capacity by the ABTS method and total phenolics; (**B1**) antioxidant capacity by the DPPH method and flavonoids, (**B2**) antioxidant capacity by the ABTS method and flavonoids; (**C1**) antioxidant capacity by the DPPH method and *ortho*-diphenols, (**C2**) antioxidant capacity by the ABTS method and *ortho*-diphenols of pods resulting from green beans grown in four distinct substrates, irrigated at 100% and 50% of WHC.

**Figure 8 plants-13-00540-f008:**
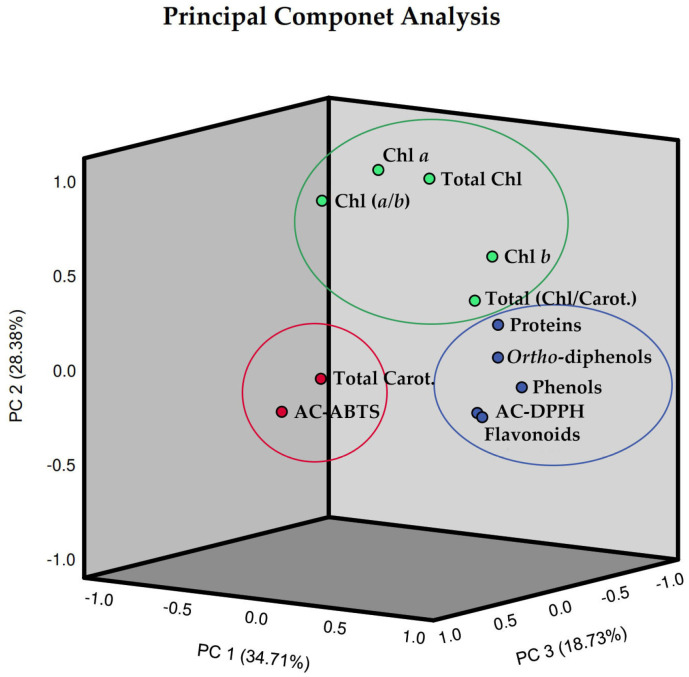
Three-dimensional Component plot, in rotated space (Varimax rotation), resulting from Principal Component Analysis (PCA) and with Kaiser Normalization, incorporating the entire dataset of pods resulting from green beans grown in four distinct substrates, irrigated at 100% and 50% of WFC. Circles showing the clustered loadings: circle blue is related to component 1, circle green is related to component 2, and circle red is related to component 3. Abbreviations: Chl *a*—chlorophyll *a*, Chl *b*—chlorophyll *b*, Chl (*a*/*b*)—chlorophyll (*a*/*b*) ratio, Total Chl—total chlorophyll (*a + b*), Total Carot.—total carotenoids, Total (Chl/Carot.)—total (Chl/Carot.) ratio, AC-ABTS—antioxidant capacity by ABTS method, AC-DPPH—antioxidant capacity by DPPH method.

**Figure 9 plants-13-00540-f009:**
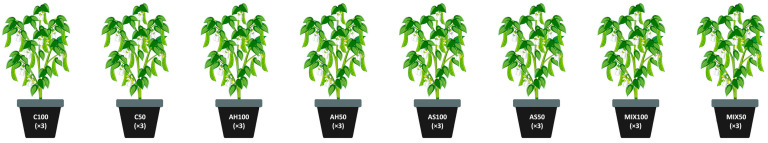
Experimental design.

**Table 1 plants-13-00540-t001:** Component scores of variables on significant principal components, eigenvalues, and the total variance explained for the dataset.

Variables	PC1	PC2	PC3
**Chl *a***	−0.069	0.983	−0.074
**Chl *b***	0.533	0.560	−0.249
**Total Chl**	0.192	0.951	−0.165
**Chl (*a*/*b*)**	−0.345	0.809	0.044
**Total Carot.**	0.267	0.055	0.928
**Total (Chl/Carot.)**	−0.080	0.160	−0.966
**Phenolics**	0.726	−0.113	−0.242
***Ortho*-diphenols**	0.884	0.125	0.201
**Flavonoids**	0.797	−0.197	0.220
**AC-DPPH**	0.767	−0.178	0.221
**AC-ABTS**	0.015	−0.146	0.924
**Proteins**	0.852	0.289	0.154
**Eigenvalues**	4.165	3.405	2.247
**Variance (%)**	34.705	28.377	18.725
**Cumulative (%)**	34.705	63.082	81.807

Extraction Method: Principal Component Analysis. Rotation Method: Varimax with Kaiser Normalization. Abbreviations: Chl *a*—chlorophyll *a*, Chl *b*—chlorophyll *b*, Chl (*a*/*b*)—chlorophyll (*a*/*b*) ratio, Total Chl—total chlorophyll (*a + b*), Total Carot.—total carotenoids, Total (Chl/Carot.)—total (Chl/Carot.) ratio, AC-ABTS—antioxidant capacity by ABTS method, AC-DPPH—antioxidant capacity by DPPH method.

## Data Availability

Data are available on request from the corresponding author.

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
