# Peer review of "Almond By-Products Substrates as Sustainable Amendments for Green Bean Cultivation"

_plants, 2024, doi:10.3390/plants13040540_

Round 1

Reviewer 1 Report

Comments and Suggestions for Authors

Reviewer 2 Report

Comments and Suggestions for Authors

The work described in this manuscript is very interesting and relevant, as it shows the potential use of almond byproducts as part of the growing medium for bean plants and the subsequent impact in pod quality. The manuscript is well written, presented, and discussed. The experiment was well designed. The data are good. In my opinion, the work is not original, as the use of almond byproducts as growing media for plants has been studied exhaustively.   The measurements are too basic. I missed HPLC work. I would recommend to modify the tittle of the work. The title should be simpler. I believe that the authors are measuring the antioxidant capacity instead of antioxidant activity.

Round 2

Reviewer 1 Report

Comments and Suggestions for Authors

Thank you for clarifications. There were an error in the numbering of the figures, and the change of the name of dots, which has now been changed. The Authors, this change can be found highlighted in green in the revised manuscript – so it was easy to found.

The most important the conclusions is improved taking into account the results of the work.